

# Quantitative analysis of the radiation error for aerial coiled fiber–optic Distributed Temperature Sensing deployments using reinforcing fabric as support structure

Armin Sigmund[1], Lena Pfister[1], Chadi Sayde[2], and Christoph K. Thomas[1]

[1]Micrometeorology Group, University of Bayreuth, Bayreuth, Germany
[2]Department of Biological and Ecological Engineering, Oregon State University, Corvallis, Oregon, USA

*Correspondence to:* A. Sigmund (armin.sigmund@uni-bayreuth.de)

**Abstract.** In recent years the spatial resolution of fiber–optic Distributed Temperature Sensing (DTS) was enhanced in various studies by helically coiling the fiber around a support structure. While solid polyvinyl chloride tubes are an appropriate support structure under water they can produce considerable errors in aerial deployments due to the radiative heating or cooling. We used meshed reinforcing fabric as a novel support structure to measure high–resolution vertical temperature profiles over

several meters height above a meadow and within and above a small lake. This study aimed at quantifying the radiation error for the coiled DTS system and the contribution caused by the novel support structure via heat conduction. A quantitative and comprehensive energy balance model is proposed and tested, which includes the shortwave radiative, longwave radiative, convective and conductive heat transfers and allows for modeling fiber temperatures as well as quantifying the radiation error. The sensitivity of the energy balance model to the conduction error caused by the reinforcing fabric is discussed in terms of

its albedo, emissivity and thermal conductivity. Modeled radiation errors amounted to -1.0 and 1.3 K at 2 m height but ranged up to 2.8 K for very high incoming shortwave radiation ($1000 \, \mathrm{J \, s^{-1} \, m^{-2}}$) and very weak winds ($0.1 \, \mathrm{m \, s^{-1}}$). After correcting for the radiation error by means of the presented energy balance the Root Mean Square Error between DTS and reference air temperatures from an aspirated resistance thermometer or an ultrasonic anemometer was 0.42 and 0.26 K above the meadow and the lake respectively. Conduction between reinforcing fabric and fiber cable had a small effect on fiber temperatures

($< 0.18 \, \mathrm{K}$). Only for locations where the fiber–optic cable attached to the reinforcing fabric touched the plastic rings supporting the fabric significant temperature artifacts of up to 2.5 K were observed. Overall, the reinforcing fabric offers several advantages over conventional support structures published to date in literature as it minimizes both radiation and conduction errors.

**Keywords.** Temperature profiles, surface layer, radiation error, turbulence, spatial observations, Distributed Temperature Sensing.

# 1 Introduction

Distributed Temperature Sensing (DTS) allows for sampling temperatures at thousands of points along a fiber–optic cable with one single instrument. Over the past 10 years Raman scatter DTS has been used in hydrology to monitor water temperatures





over both long and short distances (Selker et al., 2006; Lowry et al., 2007; Petrides et al., 2011). Recently, he DTS technique has also been applied in the atmospheric and other geo sciences: Soil moisture (Sayde et al., 2010) and wind velocities (Sayde et al., 2015) were measured by means of heated fiber–optic cables, temperature profiles in the atmospheric boundary layer (Keller et al., 2011) were obtained and turbulent structures in the surface layer could be resolved (Thomas et al., 2012). Strengths

of the DTS technique include high spatial and temporal resolutions up to 0.13 m along the fiber at 1 s (Thomas et al., 2012; Hilgersom et al., 2016b), referencing all spatially distributed measurements to a single standard, and the universal applicability in air, water, and soil.

Studying small–scale processes may require a higher spatial resolution than can be achieved with a straightly aligned fiber–optic cable, which is limited by the along–fiber resolution of the DTS instrument to several tens of centimeters. To overcome

this issue several studies have used a support structure for helically coiling the fiber–optic cable and thus enhancing the spatial resolution up to the millimeter scale in one dimension (Selker et al., 2006; Suárez et al., 2011; Vercauteren et al., 2011; Van Emmerik et al., 2013; Euser et al., 2014; Hilgersom et al., 2016a). Recently this idea was expanded to all three dimensions by spanning horizontal nets of fiber–optic cable across a cage (Hilgersom et al., 2016b).

In our application precise vertical temperature profiles over several meters height in the near–surface air layer were desired.

For this purpose, the coiled fiber–optic approach was deployed. Previous studies have pointed out two important issues for this approach: First, coiling the fiber cable around a support structure with a very small diameter (< 3.2 cm) results in temperature artifacts at the start of the coil due to significant differential attenuation caused by the strong bending (Arnon et al., 2014; Hilgersom et al., 2016a) and secondly, radiative heating of the coiled fiber–optic cable and support structure may result in substantial radiation errors up to several Kelvin particularly for aerial applications (Selker et al., 2006; Suárez et al., 2011;

Hilgersom et al., 2016a).

The radiation error for a coiled DTS system includes both the direct effect of radiative transfer to and off the fiber cable and the indirect effect of the support structure via heat conduction (here referred to as conduction error). The direct effect of radiation also applies to a straightly aligned fiber cable in open air or water. For fiber cables under water this effect is only relevant at shallow depths in clear and low–velocity water during peak solar radiation (Neilson et al., 2010). However, in aerial

DTS deployments much care has to be taken if both accurate and precise air temperatures shall be obtained. To this end de Jong et al. (2015) developed a semi–quantitative method to correct for the radiation error of fiber cables in open air using two cables with different diameters. In addition to radiative heating during sunlight hours radiative cooling at night can produce large errors in aerial deployments.

The conduction error results from differences in the radiative energy exchange of the fiber–optic cable and the support

structure. For temperature profiles under water or at the water–air interface polyvinyl chloride (PVC) tubes have been used as support structure: Suárez et al. (2011) used a white heat–shrink plastic to secure the fiber cable on a PVC tube and observed a strong influence of radiation on temperatures measured above the water surface. In order to minimize the absorption of shortwave radiation Vercauteren et al. (2011) painted the fiber–optic cable and the PVC tube with white antifouling marine paint and installed white plastic disks for shading.





To date, a comprehensive and quantitative analyses of the conduction error for coiled DTS systems is lacking. For a per-forated PVC tube Hilgersom et al. (2016a) provided semi–quantitative estimates for the conduction error above water by subtracting the temperature of fiber sections crossing the perforations from the temperature of fiber sections attached to the tube. These rough estimates yielded conduction errors between -0.4 and 0.1 K and up to 0.7 K during a fog event. Using a

simple energy balance model Hilgersom et al. (2016a) could provide some physical explanation for the observed temperature differences by the effect of the PVC tube. However, accurate estimates for the conduction error were not reported as the di-ameter of the tube perforations was much smaller than the spatial resolution of the DTS instrument and the model deployed a simplified shortwave radiative transfer scheme for cloudy skies (Hilgersom et al., 2016a).

In order to avoid the large conduction errors found for a solid PVC tube Van Emmerik et al. (2013) developed an open hyper-

boloid PVC frame minimizing the contact area and thus the conduction error between fiber cable and support structure. Similar to this idea, our study presents another novel and inexpensive support structure composed of a white, meshed reinforcing fabric and several transparent acrylic glass rings. Sayde et al. (2015) proposed a detailed and quantitative energy balance model for an actively heated fiber cable in open air for the purpose of measuring the wind velocity orthogonal to the fiber–optic cable. This model included terms for individual shortwave and longwave radiation fluxes and convective heat transfer. In this study

we adapted this energy balance model of Sayde et al. (2015) to the coiled DTS system by including heat conduction between fiber cable and reinforcing fabric and the influence of the new fiber geometry on the interception of direct radiation.

The overarching objective of our study was to quantify the radiation error for this novel setup of the aerial DTS deployment by means of a comprehensive and quantitative energy balance model. The three specific objectives were to (1) quantify and if possible correct for the radiation error of the aerial coiled DTS deployment by means of a full energy balance model validated

with measurements, (2) estimate artifacts caused by the novel support structure via conduction and (3) visualize the utility of coiled–fiber deployments for observing high–resolution air temperature profiles near the surface ground.

## 2   Materials and methods

### 2.1   Theory of Raman scatter DTS

The DTS technique allows simultaneous measurements at thousands of points in space by injecting laser pulses into a fiber–

optic cable. Raman scattering systems measure the intensities of non–elastic backscatter having slightly lower (Stokes signal) or higher (anti–Stokes signal) frequencies than the original laser light (Selker et al., 2006). For low light intensities both the Stokes and the anti–Stokes signals depend on the magnitude of illumination but only the anti–Stokes signal additionally depends on fiber temperature (Selker et al., 2006). Therefore the anti–Stokes to Stokes ratio can be used to infer the relative fiber temperature which should be close to the temperature of the surrounding medium. The point of measurement is determined

by the time of travel of the light (Selker et al., 2006). The precision of Raman DTS measurements decreases for smaller spatial or shorter temporal averaging and exponentially decreases with the cable length between the instrument and the point of measurement. Even though there has been progress toward increasing the maximum spatial resolution to tens of centimeters



along the fiber, the maximum attainable resolution is limited by the finite signal strengths, strong optical dispersion, and signal loss due to trimming the light pulse (Selker et al., 2006; Tyler et al., 2009).

## 2.2 Coiling the fiber around reinforcing fabric

In our experiment the instrument–specific along–fiber averaging was limited to 1 m. By helically coiling the fiber–optic cable
around a reinforcing fabric column the vertical resolution was enhanced to the centimeter scale. In a field experiment two hollow columns were located on a meadow (meadow–column) and in a nearby small lake (lake–column) with 3.0 and 5.1 m height respectively (Fig. 1). Both columns had a circumference of approximately 1.00 m which was equal to the along–fiber resolution of the DTS instrument. In order to achieve the highest vertical resolution of air temperature and its gradients close to the surface ground the individual fiber coils were spaced at 1 cm vertical separation distance in the lower part of the columns
(Fig. 1a,c) and at 5 cm in the upper parts.

    The columns consisted of white reinforcing fabric with a mesh size of approximately 3.8 mm. This material is composed of glass fibers and is able to minimize the radiation error for the following reasons. Since both the fabric skeins and the fiber–cable (0.9 mm diameter, AFL, Mönchengladbach, NW, Germany) were white and had a similar diameter radiative heating or cooling were expected to be small and similar. A blue lettering printed onto the fabric was removed using cotton balls soaked in alcohol
in order to ensure the high albedo of the fabric. The high air permeability of the meshed fabric was expected to allow for strong convective heat exchange between the fiber–optic cable and the air reducing the temperature differences between fabric, fiber, and the air. Furthermore, the contact area between the meshed fabric and the fiber cable was reduced to a fraction compared to conventional solid constructions using PVC pipes. The thermal conductivity of the fabric was expected to be low compared to that of other materials.

The reinforcing fabric was formed into a column by hot glueing with an approximately 12 cm wide overlap. This fabric column was hot–glued to transparent acrylic glass rings spaced at 1 m intervals for mechanical stabilization. The acrylic glass rings were attached to a metal pole in the center of each column (Fig. 1). The contact area between fabric and acrylic glass ring was 2.8 cm wide. The optical fiber was then attached to the outer face of the reinforcing fabric column with four dots of hot glue per 1 m long coil.

## 25  2.3  Field experiment

The high–resolution profile measurements were part of the Cold Air Drainage Experiment (CADEX) conducted from 13 March 2015 to 29 April 2015 in the Ecological Botanical Gardens of the University of Bayreuth, Bayreuth, Germany (44°55′27.48″ N, 11°35′6.36″ E). A single fiber–optic cable of approximately 2 km length allowed distributed temperature measurements along a gently inclined meadow (approximately 1.3 °) and a small lake. The main objective of CADEX was to investigate the
dynamics of cold–air drainage and pooling. In this paper we confine ourselves to analyzing radiation effects on the coiled DTS system used to capture centimeter–resolution temperature profiles over several meters height.

    The meadow–column was located on a gentle slope within a few meters of a long–term weather and climate station, which provided the reference observations for direct and total incoming shortwave irradiance (pyranometer SDE, type 9.1, UTK –



EcoSens GmbH, Zeitz, ST, Germany), incoming and outgoing longwave irradiance (pyrgeometer CG2, Kipp & Zonen B.V., Delft, ZH, The Netherlands), dry bulb temperature at 2 m height (electrically aspirated psychrometer by Frankenberger, no. 3010.0000, Theodor Friedrichs & Co., Schenefeld, SH, Germany) as well as wind velocity and direction at 17 m height (Ultrasonic Anemometer 2D, Adolf Thies GmbH & Co. KG, Göttingen, NI, Germany).

The lake–column was horizontally separated from the meadow–column by 87 m and provided measurements of water and air temperatures. Both columns were sampled by the same DTS instrument (ORYX DTS, Sensornet Ltd, Elstree, HERTS, United Kingdom) with a frequency of 8.5 kHz. Next to the lake–column an ultrasonic anemometer (CSAT3, Campbell Sci., Logan, UT, USA) and an open–path infrared gas analyzer (LI 7500, LI–COR, Lincoln, NE, USA) were installed at 2.13 m height and provided wind velocity and dry–bulb reference air temperature used to quantify radiation and conduction errors of
the coiled DTS system above the lake.

## 2.4   DTS sampling

The fiber–optic cable was attached to the DTS instrument in a double–ended configuration but the instrument was operated in single–ended mode, i.e. measurements of the two directions were stored separately. The along–fiber averaging was 1 m. An appropriate averaging time was determined in a previous laboratory experiment:

Two coiled fiber sections of approximately 50 m length were inserted in a mechanically stirred cold water bath of known temperature and sampled with varying averaging times (Fig. 2a). As expected the spatial standard deviation was approximately proportional to the term $\sqrt{n}^{-1}$ (Fig. 2b), where $n$ is the number of samples per averaging time. For the field experiment an averaging time of 30 s was chosen allowing to capture short–lived temperature changes while keeping the random sampling error small at approximately 0.04 K in the laboratory.

## 2.5   Analytical methods

### 2.5.1   Post–field data processing

The temperature output of the DTS instrument was calibrated based on two thermally insulated baths with heated and ice–cooled water. Approximately 50 m of coiled fiber passed through the baths at both ends. The water was stirred by an aquarium pump and the reference temperature was measured with a Pt100 sensor sampled by the DTS instrument. The post–field calibra-
tion was performed dynamically in two steps: At first the DTS temperatures were offset–corrected by means of the cold bath. The offset was calculated as a linear function of length along the fiber given by the mean offsets between DTS and reference temperatures at each cable end. In the second step the temperatures were slope–corrected with a linear function of temperature given by the mean DTS and reference temperatures of both baths.

The post–processing also included the transformation of the measurement positions from length along the fiber into height
above surface. For mapping purposes individual sections of the fiber were cooled at the beginning and end of each column, while the positions in between were inferred by means of the column proportions. Results were cross–checked against the counted fiber coils. The accuracy of the positions was approximately ±2 m along the fiber corresponding to ±2 cm in height.



The acrylic glass rings of the support structure caused obvious artifacts due to solar heating for daytime and radiative cooling at night, especially at lower heights and weak winds (Fig. 3a,b). The temperature anomalies were up to 2.5 K at particular points in time (not shown). These artifacts were removed during post–processing except in the water where artifacts could be determined visually. On average 4 measurement values were removed per acrylic glass ring. Removed data were replaced by
linear inter– and extrapolation based on the nearest 10 data points.

As the signal–to–noise ratio decreases with length along the fiber (Selker et al., 2006) the precision of the temperature profiles changed with measurement direction (every 30 s) and location (meadow, lake). In order to avoid this heteroscedasticity subsequent measurements with alternating directions were aggregated to 1 min averages.

For the averaged temperature profiles some random scatter remained. This uncertainty can be quantified as the spatial
standard deviation of DTS measurements in the calibration baths with a mean error of 0.12 K. In order to remove this random scatter in the temperature profiles a spatial wavelet filter based on the biorthogonal set of wavelets BIOR5.5 was applied (Fig. 3c) (Thomas and Foken, 2005). Because of the different vertical resolutions the lower and upper profile sections were filtered separately with critical lengths of 2.4 and 11.8 cm respectively. The section with a strong temperature discontinuity at the water–air interface (from -6 to 0 cm height) was not filtered in order to avoid unrealistic deviations between filtered and
unfiltered profiles. Instead this section was interpolated according to Akima (1970) at 0.1 cm increments. For both the meadow and the lake location the temporal mean of the Root Mean Square Error between filtered and unfiltered profiles was 0.08 K and thus accounted for a major fraction of the mean random sampling error of 0.12 K, which demonstrated the effectiveness of the applied wavelet filter.

### 2.5.2   Comprehensive energy balance model

Radiation errors for the coiled DTS system were quantified by modeling the fiber temperature via an energy balance approach. The model included the contribution of heat conduction from the support structure to the radiation error and can be applied to a wide range of meteorologic conditions or material properties. Since the DTS measurements were averaged over one fiber–optic winding the energy balance model was proposed for the same spatial extent. The sum $Q$ of energy fluxes to and off a fiber–optic winding can be expressed as (adapted from Sayde et al. , 2015):

$$Q = Q_{in}^{cond} + Q_{in}^{S} + Q_{in}^{L} - Q_{out}^{L} - Q_{out}^{conv} \tag{1}$$

where $Q_{in}^{cond}$ is the incoming energy flux from conduction between fiber cable and reinforcing fabric, $Q_{in}^{S}$ and $Q_{in}^{L}$ are the incoming energy fluxes from shortwave and longwave radiation respectively, $Q_{out}^{L}$ is the outgoing fiber's emittance of longwave radiation and $Q_{out}^{conv}$ is the outgoing turbulent convective energy exchange with the moving air (Fig. 4a). The unit of all terms is $J\,s^{-1}$. Incoming fluxes are positive when directed to the fiber–optic cable while outgoing fluxes are positive when directed
off the cable. On sunny days a fiber–optic winding was expected to be warmer on the sunlit side compared to the shaded side of the column. Conduction along the fiber between coils was neglected as it is assumed that the energy transfer into/ from the sunlit/shaded ends of one fiber coil into the adjacent are equal in magnitude but opposite in sign and thus cancel out. Assuming that the cross section of the fiber cable has a homogeneous temperature the sum $Q$ of incoming and outgoing energy fluxes





corresponds to the rate change of internal energy stored in one fiber–optic winding:

$$Q = c_p \rho V \frac{\mathrm{d}T_s}{\mathrm{d}t} \qquad (2)$$

where $c_p$ [J kg$^{-1}$ K$^{-1}$] is specific heat capacity of the fiber cable, $\rho$ [kg m$^{-3}$] is density of the fiber cable, $V$ [m$^3$] is volume of one fiber coil, t is time [s] and $T_s$ [K] is the fiber's surface temperature which is assumed to be homogeneous and equivalent to

the temperature inside the fiber cable.

The energy flux from conduction $Q_{in}^{cond}$ [J s$^{-1}$] between fiber cable and reinforcing fabric was calculated on the basis of Fourier's law (Çengel, 1998):

$$Q_{in}^{cond} = kA\frac{(T_r - T_s)}{\Delta x} \qquad (3)$$

This is a simplification as conduction is assumed to be one–dimensional similar to a wall with thermal conductivity $k$

[J s$^{-1}$ m$^{-1}$ K$^{-1}$], cross–sectional area $A$ [m$^2$] (contact area between one fiber coil and reinforcing fabric) and thickness $\Delta x$ [m] (distance between the centers of the fiber and the touching skein of reinforcing fabric). The fiber's surface temperature $T_s$ [K] and reinforcing fabric temperature $T_r$ [K] are to be homogeneous for each winding. We approximated the slightly lenticular–shaped cross section of the reinforcing fabric skeins by a cylinder with a radius of 0.45 mm, which is identical to that of the fiber–optic cable.

Since $T_r$ was initially unknown the model was solved iteratively in two steps (Fig. 4b): First, $T_r$ and $T_s$ were modeled via separate energy balances without the unknown conduction term. In the second step these temperatures were used to include conduction in the fiber's balance equation and to solve again for $T_s$. Modeled fiber temperatures were validated by means of the DTS temperatures. Finally, conduction errors and total radiation errors were determined as $T_{s1} - T_{s0}$ and $T_{s1} - T_A$ respectively where the indices 0 and 1 refer to models without and with conduction respectively (Fig. 4b).

We adopted the temporal resolution of the input data from the long–term weather station of 10 min means for our model. In case of the lake–column, the sonic temperatures over the lake were converted into dry–bulb temperatures using the mean water vapor density measured by the open–path gas analyzer. Reference temperatures for both locations were corrected for an instrument–specific offset amounting to -0.377 and -0.087 K over the meadow and the lake respectively by evaluating a linear model for nighttime data with small shortwave irradiance (< 50 J s$^{-1}$ m$^{-2}$) and high wind velocities (> 2 m s$^{-1}$ at 2.13 m

height). Since the offset was independent of temperature the slope of the fitted line was set 1.

Combining Eq. (1) and (2) and including conduction (Eq. (3)) gives rise to the following energy balance of one fiber–optic winding:

$$\frac{1}{2}c_p \rho r \frac{\mathrm{d}T_s}{\mathrm{d}t} = \frac{1}{2\pi r B} kA\frac{T_{r0} - T_{s0}}{2r} + (\bar{S}_b + \bar{S}_d + \rho\bar{S}_t)(1-a) + (\bar{L}_\downarrow + \bar{L}_\uparrow)\varepsilon - \varepsilon\sigma T_{s1}^4 - h(T_{s1} - T_A) \qquad (4)$$

where $r$ [m] is fiber cable radius, $B$ [m] is the length of the fiber winding, $\bar{S}_b$, $\bar{S}_d$, and $\rho\bar{S}_t$ [J s$^{-1}$ m$^{-2}$] are the means of

direct, diffuse, and reflected shortwave radiation fluxes over unit surface area of the fiber–optic winding respectively, $\bar{L}_\downarrow$ and $\bar{L}_\uparrow$ [J s$^{-1}$ m$^{-2}$] are the means of downward and upward longwave radiation fluxes over unit surface area of the fiber–optic winding respectively (Fig. 4a), $a$ [1] is fiber cable albedo, $\varepsilon$ [1] is fiber cable emissivity, $\sigma$ [J s$^{-1}$ m$^{-2}$ K$^{-4}$] is the Stefan–Boltzmann constant, $h$ [J s$^{-1}$ m$^{-2}$ K$^{-1}$] is the convection heat–transfer coefficient and $T_A$ [K] is reference air temperature.



The energy balance of one horizontal skein of reinforcing fabric is the same as for the fiber cable (Eq. (4)) when replacing specific heat capacity, density and emissivity by the correspondent values of the reinforcing fabric and exchanging fiber and fabric temperatures. The temperature tendencies $\mathrm{d}T_s\,\mathrm{d}t^{-1}$ $[\mathrm{K\,s^{-1}}]$ of fiber and fabric were both approximated by approximate finite differences using the first and last 1min DTS temperatures within each 10min interval. For the lake–column the reflected

shortwave irradiance was estimated by means of the albedo of water and the upward longwave irradiance was estimated according to the Stefan–Boltzmann law using the DTS temperature measured directly below water surface. Assuming that diffuse and reflected shortwave radiation as well as down– and upward longwave radiation are omnidirectional the interception of these fluxes by a fiber–optic winding can be expressed as (Monteith and Unsworth, 2008):

$$\bar{X} = \frac{X_h}{2} \tag{5}$$

where $\bar{X}$ $[\mathrm{J\,s^{-1}\,m^{-2}}]$ is $\bar{S}_d$, $\rho\bar{S}_t$, $\bar{L}_\downarrow$ or $\bar{L}_\uparrow$ and $X_h$ $[\mathrm{J\,s^{-1}\,m^{-2}}]$ is the correspondent flux relative to a horizontal plane. Interception of direct shortwave radiation was based on the following equation for a horizontal cylinder (Monteith and Unsworth, 2008):

$$\bar{S}_{b,cyl} = \frac{\operatorname{cosec}\beta\,(1 - \cos^2\beta\cos^2\theta)^{0.5}}{\pi} S_{b,h} \tag{6}$$

where $\bar{S}_{b,cyl}$ $[\mathrm{J\,s^{-1}\,m^{-2}}]$ is the mean direct radiation over unit cylinder surface area, $S_{b,h}$ $[\mathrm{J\,s^{-1}\,m^{-2}}]$ is direct radiation

measured on a horizontal plane, $\theta$ [°] is the angle between cylinder axis and solar azimuth and $\beta$ [°] is sun elevation which was calculated from the geographic coordinates and time by means of a flux data processing software. A fiber coil can be approximated by several cylinder segments with varying orientation, i.e. varying angles $\theta$. Hence the mean direct radiation $\bar{S}_b$ $[\mathrm{J\,s^{-1}\,m^{-2}}]$ intercepted by a fiber coil was calculated by averaging $\bar{S}_{b,cyl}$ for 19 different angles $\theta$:

$$\bar{S}_b = \frac{1}{10} \sum_{i=0}^{18} \bar{S}_{b,cyl}(\theta = 5° \cdot i) \tag{7}$$

A number of 19 angles $\theta$ was deemed a sufficient approximation as a doubling of this number would change $\bar{S}_b$ by less than 0.6 %.

The convective heat–transfer coefficient $h$ from Eq. (4) mainly depends on wind velocity and can be parameterized as (Sayde et al., 2015; Lundström et al., 2007; Zkauskas, 1987):

$$h = C\,(2r)^{m-1}\,\mathrm{Pr}^n \left(\frac{\mathrm{Pr}}{\mathrm{Pr}_s}\right)^{\frac{1}{4}} K_A\,\upsilon_A^{-m}\,U_N^m \tag{8}$$

where $r$ [m] is fiber cable radius, $K_A$ $[\mathrm{J\,s^{-1}\,m^{-1}\,K^{-1}}]$ is thermal conductivity of air and $\upsilon_A$ $[\mathrm{m^2\,s^{-1}}]$ is kinematic viscosity of air. The dimensionless parameters $C$ and $m$ were set 0.52 and 0.5 respectively for Reynolds numbers from 40 to 1000 and they were set 0.75 and 0.4 respectively for Reynolds numbers between 1 and 40 (Zkauskas, 1987). Pr and $\mathrm{Pr}_s$ [1] are Prandtl numbers at air and fiber temperature respectively. The Prandtl numbers are given by $\upsilon\alpha^{-1}$ and were calculated by linear inter– and extrapolation based on Table 1 (Bejan, 2004). The term $(\mathrm{Pr}\,\mathrm{Pr}_s^{-1})^{\frac{1}{4}}$ in Eq. (8) was set 1 because even for fiber and air

temperatures of 20 and 0 °C respectively it would be 0.9987. $n$ is another dimensionless parameter set to 0.37 since all Pr





were less than 10 (Zkauskas, 1987). For a straightly oriented fiber only the wind velocity component $U_N$ $[\mathrm{m\,s^{-1}}]$ normal to the fiber cable contributes to convection. In contrast, in case of a coiled fiber the total three–dimensional wind velocity can be used instead of $U_N$ because of its different geometry. The wind velocity at the meadow–column was assumed to be identical over the lake.

Material properties used to model the fiber's energy balance are listed in Table 2. Since the thermal conductivity $k$ was uncertain the model was evaluated for a lower and upper boundary of $k$. The albedos of fiber and fabric were unknown but we assumed them to be equal to a typical cloud albedo because of their white color. Conduction between fiber and fabric depends on differences in albedo and/or emissivity. Therefore the sensitivity of the conduction error was investigated by varying the fabric albedo and emissivity (Table 2).

The energy balance model can be used to correct for the radiation error of the coiled DTS system even if reference air temperatures were not measured. In this case the DTS temperature $T_f$ is inserted as $T_s$ in Eq. (4) to solve for the air temperature. As will be shown in the subsequent section the conduction term can be neglected for this experimental setup and thus the corrected fiber temperatures $T_{Af}$ [K], which are an estimate for the air temperature, can be calculated according to the following correction model:

$$15 \quad T_{Af} = T_f + \frac{1}{h}\left(\frac{1}{2}c_p\rho r\frac{\mathrm{d}T_s}{\mathrm{d}t} - (\bar{S}_b + \bar{S}_d + \rho\bar{S}_t)(1-a) - (\bar{L}_\downarrow + \bar{L}_\uparrow)\varepsilon + \varepsilon\sigma T_f^4\right) \tag{9}$$

where $h$ can be approximated using the observed fiber temperature instead of the unknown air temperature.

## 3   Results and discussion

### 3.1   Evaluation of the radiation error and artifacts caused by the reinforcing fabric

Prior to analyzing the radiation error the energy balance model was validated by comparing its estimates with the temperature
measurements in the 5–day–period from 6 to 10 April 2015. Daily maximum incoming shortwave irradiance ranged from 677 to 839 $\mathrm{J\,s^{-1}\,m^{-2}}$. On 6 and 8 April the sky was partly cloud–covered whereas the other days were predominately clear–sky, particularly on 10 April. Daytime wind speeds varied between 6 April (mean: 1.5 $\mathrm{m\,s^{-1}}$, maximum: 3.2 $\mathrm{m\,s^{-1}}$ at 2.13 m height above the lake) and other days (mean: 0.7 $\mathrm{m\,s^{-1}}$, maximum: 2.0 $\mathrm{m\,s^{-1}}$). Nighttime wind speeds were generally small (mean: 0.2 $\mathrm{m\,s^{-1}}$, maximum: 1.5 $\mathrm{m\,s^{-1}}$). Relatively large radiation errors were expected due to the light winds and intense
solar radiation.

The Root Mean Square Error (RMSE) for the modeled and measured temperatures was insensitive to the choice of including or excluding the conduction term (Table 3). The RMSE equaled 0.40 and 0.25 K above the meadow and the lake, respectively. Including the conduction term in the model did not reduce the RMSE between model and measurement (Table 3). This suggests that conduction between reinforcing fabric and fiber cable was negligible given the initial model parameters. Hence the
conduction term was omitted for our experimental configuration when correcting the DTS temperatures for the radiation error. A comparison between the RMSE for modeled and measured air temperature and that between the modeled and measured fiber





temperatures did not yield significant differences (Table 3). The slightly lower performance could be attributed to evaluating the convective heat–transfer coefficient at the fiber instead of the air temperature.

Correcting DTS temperatures for the radiation error reduced the RMSE between DTS and reference temperature above the meadow by 41 % from 0.71 to 0.42 K (Table 3). Our results can be compared with those from the approach of de Jong

et al. (2015) for correcting radiation effects by extrapolating the temperature of two fiber cables with different diameters to a zero diameter under the assumption of forced convection. de Jong et al. (2015) also measured above a grassland over a similar 5–day–period but with linearly and horizontally aligned fiber cables in open air. They reported an RMSE for the uncorrected measurements of 0.74 and 0.61 K comparing white cables of 3.0 and 1.6 mm diameter respectively with their reference measurements. After correction they reported an RMSE of 0.38 K representing a similar agreement as achieved

by our energy balance approach in this study. In contrast to the method of de Jong et al. (2015) the comprehensive energy balance approach proposed here requires additional radiation and wind measurements. However, the energy balance approach offers the advantage of being applicable also for weak wind conditions when shear–induced forced convection is small or absent and is able to include errors resulting from the support structure. Above the lake the energy balance model reduced the RMSE between DTS and reference temperature by 60 % from 0.65 to 0.26 K (Table 3). The mismatch between model and

measurements was larger above the meadow than above the lake, which could result from assuming an identical wind velocity at both locations.

The temporal course of temperatures, radiation error and uncertainties of selected model versions from 8 to 10 April are presented in Fig. 5. The model was able to quantitatively reproduce the heating and cooling of the fiber compared to the air during the sunlight and dark hours respectively (Fig. 5c,d). The modeling uncertainty quantified by subtracting the measured

from the modeled fiber temperatures was insensitive to including the conduction term in the model (Fig. 5e,f), which underlines the utility of choosing reinforcing fabric as support structure for the fiber. Over the meadow the modeling uncertainty varied around a mean of 0.3 K, while over the lake the mean was approximately equal to 0.0 K (Fig. 5e,f). This observation is in agreement with the higher RMSE observed over the meadow than over the lake (Table 3). The standard deviation of the modeling uncertainty was similar above the meadow (0.28 K) and the lake (0.26 K). Corrected fiber temperatures were almost

identical to reference air temperatures, which validates the proposed energy balance model (Fig. 5e–h).

The modeled radiation error ranged between -1.0 and 1.3 K and -1.0 and 1.0 K above the meadow and the lake respectively (Fig. 5c,d). Hence the radiative cooling at night is of similar magnitude than the daytime heating for our setup. The heating effect of a higher shortwave radiation during the day are to a large degree compensated by the effect of higher wind speeds. Hence, maximum radiation errors during the day can be expected for clear–skies and weak winds. The reported results are

representative for typical measurement heights at 2 m above the surface ground. Radiation errors closer to the ground are expected to increase due to the weaker winds and resulting less efficient convective heat transfer. If measurements were taken in summer or at lower latitude the incoming shortwave irradiance could be higher. The radiation error was modeled for all possible combinations of the two main drivers incoming shortwave radiation and wind speed (Fig. 6) over the meadow based on conditions from 10 April, 12:00 local standard time (LST). Air temperature (16.8 °C), sun elevation (53 °), the ratio of

direct to total incoming shortwave irradiance ($S_b \, S_t^{-1} = 0.91$) and net longwave radiation (117 J s$^{-1}$ m$^{-2}$) were kept constant





while $S_t$ and $U$ were varied (initially 724 $\mathrm{J\,s^{-1}\,m^{-2}}$ and 0.26 $\mathrm{m\,s^{-1}}$ respectively). As expected, the radiation error increased strongly with $S_t$ and decreased with $U$. For typical daytime conditions in the measurement period ($S_t < 700$ $\mathrm{J\,s^{-1}\,m^{-2}}$, $U > 1\,\mathrm{m\,s^{-1}}$) radiation errors at 2.0 m height were generally less than 0.8 K (Fig. 6). At high $S_t$ the influence of $U$ on the radiation error was strong. This suggests that radiation errors close to the ground can be substantially higher than at 2 m height. Under

extreme conditions ($S_t = 1000$ $\mathrm{J\,s^{-1}\,m^{-2}}$, $U = 0.1$ $\mathrm{m\,s^{-1}}$) the model predicted a radiation error of 2.8 K (Fig. 6). However, even these extreme radiation errors can be corrected for using the proposed energy balance model if all input variables are known. Since $S_t$ varies with sun elevation and the radiation error is sensitive to the ratio of direct to total solar irradiance ($S_b\,S_t^{-1}$) Fig. 6 represents an approximation to actual conditions. If the sun elevation was 75 $^{\circ}$ instead of 53 $^{\circ}$ the radiation error under extreme conditions ($S_t = 1000$ $\mathrm{J\,s^{-1}\,m^{-2}}$, $U = 0.1$ $\mathrm{m\,s^{-1}}$) would be reduced by 0.2 K. In case of cloud cover

resulting in $S_t = 400$ $\mathrm{J\,s^{-1}\,m^{-2}}$ and $S_b\,S_t^{-1} = 0.2$ instead of 0.91 the radiation error at $U = 0.1$ $\mathrm{m\,s^{-1}}$ would increase by 0.3 K since the diffuse radiation is omnidirectional and therefore more efficient in heating the fiber than direct–beam radiation.

    The sensitivity of the conduction error to differences in albedo and emissivity between fiber cable and reinforcing fabric was investigated for 2 m height above the meadow on 10 April 2015, 12:00 LST (Fig. 7). Albedo and emissivity of the fiber cable were kept constant while the fabric's albedo was varied by maximum ±20 % and the fabric emissivity by maximum -12

and +33 %. Assuming that thermal conductivity of fabric and fiber was low (0.1 $\mathrm{J\,s^{-1}\,m^{-1}\,K^{-1}}$) the conduction error was maximum ±0.03 K across the entire range of albedo and emissivity (Fig. 7a). In this case, the conduction error accounted for less than 2 % of the radiation error and was still deemed negligible. For a high thermal conductivity (0.6 $\mathrm{J\,s^{-1}\,m^{-1}\,K^{-1}}$) conduction errors are significant and increased to -0.16 K for a large albedo of 0.84 and an emissivity of unity and to 0.18 K for a small fabric albedo of 0.56 and an emissivity of 0.66 (Fig. 7b). Here the contribution of conduction to the total radiation

error increases to 12 %. In general, the conduction error must be accounted for if thermal conductivity and the difference in albedos are high (Fig. 7). However, we found that the maximum modeled conduction error was always smaller than temperature artifacts caused by the acrylic glass rings supporting the reinforcing fabric, which amounted to approximately 2.5 K. For a fiber cable coiled around a perforated PVC tube Hilgersom et al. (2016a) reported a mean measured conduction error of -0.4 and 0.1 K with extremes up to 0.7 K during a fog event. One must recall that in this study the conduction error was defined as

the difference between the fiber temperature for open air and fiber temperature at the perforated PVC tube. In comparison, the reinforcing fabric applied in our experiment caused much smaller conduction errors.

### 3.2   Observed high–resolution temperature profiles

To illustrate the utility of the coiled fiber deployment for observing high–resolution vertical temperature profiles close to the ground we selected a night with clear skies (Fig. 8). The night was characterized by a significant longwave radiative cooling

of 64 $\mathrm{J\,s^{-1}\,m^{-2}}$ and weak winds ranging between 0.03 and 0.98 $\mathrm{m\,s^{-1}}$ at 2.13 m height above the lake between 18:00 and 22:00 LST. The temperature profiles are presented as deviations from the top of the column for the purpose of removing the diurnal cycle and highlighting its temporal variability in the vertical. The night shows a strongly static stability over the meadow (Fig. 8a,b), which is maximum for the period 18:00 and 22:00 LST and amounting to -5.5 K. We observed a strong temporal variability of these vertical differences indicated by the blue streaks of cooler air (Fig. 8a). Over the course of the





night the static stratification became diminished while the temporal variability was maintained. Above the lake statically stable conditions with a vertical temperature difference of up to -3.4 K occurred only until 21:00 LST. As the night continued, the air became statically unstable in the lowest 0.5 m above water and isothermal further aloft (Fig. 8c,d). Close to the water surface the air was warmer by maximum 1.9 K than at the profile top due to a higher heat capacity of the water causing an upward

directed sensible heat flux. The temporal variability was similarly strong compared to the meadow, indicative of a vigorous turbulent transport despite the stable or neutral stratification.

## 4   Conclusions

In this study we quantified the radiation error for a helically coiled fiber–optic cable around a novel support structure observed using distributed temperature sensing by proposing a comprehensive energy balance model, which includes terms for shortwave

radiative, longwave radiative, convective, and conductive heat transfers. With regard to the three objectives defined in the introduction, we arrive at the following conclusions:

In the investigated period the modeled radiation error ranged from -1.0 to 1.3 K and from -1.0 to 1.0 K at 2 m height above the meadow and the small lake respectively. For very high incoming shortwave irradiances ($1000 \, \mathrm{J\,s^{-1}\,m^{-2}}$) and very weak winds ($0.1 \, \mathrm{m\,s^{-1}}$) the fiber cable was predicted to be 2.8 K warmer than the air. If accurate air temperature profiles are desired

we strongly recommend correcting the DTS temperatures for the radiation error using the presented energy balance model, which requires measuring or estimating the wind velocity profile and the shortwave and longwave radiation components. The proposed model was validated for a broad range of atmospheric conditions. Compared to existing approaches available from the DTS literature, it offers the advantage of being applicable also for weak winds and close to the surface ground. The correction reduced the RMSE between DTS and reference measurements by 41 and 60 % resulting in an absolute RMSE of 0.42 and

0.26 K above the meadow and the lake respectively. We recommend measuring the wind speed in–situ in close proximity to the coiled optical fiber because of the sensitivity of the model to the convective cooling.

The reinforcing fabric is an excellent support structure for aerial DTS deployments since its artifacts on the observed fiber temperature via conduction are very small or negligible. The model results suggest that the conduction term can be neglected in the fiber's energy balance as long as the thermal conductivity of the material is small and the difference in albedos between

the fiber and the fabric are negligible. In the worst possible case the conduction error increases to 13 % proportional to the total radiation error. The structures supporting the reinforcing fabric, which in our case was done via acrylic glass rings spaced at 1 m intervals, caused significant conduction errors where the ring touched the fabric of up to 2.5 K. Therefore the number and dimensions of these rings should be reduced to a minimum.

For a clear night with weak winds the measured temperature profiles showed sharp vertical gradients, varied strongly in

time, and revealed vertical temperature differences of up to -5.5 K between the meadow surface and the top of the profile at 3 m height. The higher vertical resolution of the coiled–fiber deployment therefore allows for more realistic observations of the natural spatiotemporal variability of the air temperature field near the surface ground.





## 5 Data availability

Data used in the analysis will be provided on request. Please contact the authors.

*Competing interests.* The authors declare that they have no conflict of interest.

*Acknowledgements.* This research was partially supported by the US National Science Foundation CAREER award AGS 0955444. The

5 fiber–optics instrument was provided by the Center for Transformative Environmental Monitoring Programs (CTEMPS) funded by the US

National Science Foundation, award EAR 0930061. We thank Prof. Dr. John Selker for technical support and Dr. Wolfgang Babel and

Johannes Olesch for technical assistance in the field experiment. We also thank Dr. Gregor Aas who placed the measurement site in the

Ecological Botanical Gardens at the disposal.



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





**Table 1.** Properties of dry air at atmospheric pressure (Bejan, 2004). $\upsilon$ is kinematic viscosity, $K$ is thermal conductivity and $\alpha$ is thermal diffusivity.

| Temperature [°C] | $\upsilon$ [cm$^2$ s$^{-1}$] | $K$ [J s$^{-1}$ m$^{-1}$ K$^{-1}$] | $\alpha$ [cm$^2$ s$^{-1}$] |
|---|---|---|---|
| 0 | 0.132 | 0.024 | 0.184 |
| 10 | 0.141 | 0.025 | 0.196 |
| 20 | 0.150 | 0.025 | 0.208 |
| 30 | 0.160 | 0.026 | 0.233 |





**Table 2.** Material properties used in the energy balance model. A fiber cable coil and a reinforcing fabric skein are briefly called fiber and fabric respectively. Numbers in brackets express the range used for sensitivity analysis. $T$ denotes temperature.

| Symbol | Description | Values | Units | Explanation |
|---|---|---|---|---|
| $c_p$ | Specific heat capacity of fiber | 1.06 | $\mathrm{kJ\,kg^{-1}\,K^{-1}}$ | mean for PVC[1], Kevlar fiber[2] and glass[3] |
| $c_{pr}$ | Specific heat capacity of fabric | 0.80 | $\mathrm{kJ\,kg^{-1}\,K^{-1}}$ | as for A–glass fiber[4] |
| $\rho$ | Density of fiber | 1.8 | $\mathrm{g\,cm^{-3}}$ | mean for PVC[5], Kevlar fiber[2] and E–glass[6] |
| $\rho_r$ | Density of fabric | 2.4 | $\mathrm{g\,cm^{-3}}$ | as for A–glass fiber[4] |
| $r$ | Radius of fiber or fabric | 0.45 | mm | |
| $B$ | Length of fiber or fabric | 1.01 | m | |
| $k$ | Thermal conductivity for fiber and fabric | low: 0.1 high: 0.6 | $\mathrm{J\,s^{-1}\,m^{-1}\,K^{-1}}$ | mean for PVC[5] (fiber jacket) and fiberglass/ glass[7] (fabric) |
| $A$ | Contact area of fiber and fabric | 0.5 | $\mathrm{cm^2}$ | estimated |
| $a$ | Albedo of fiber | 0.7 | 1 | estimated as for clouds[8] |
| $a_r$ | Albedo of fabric | 0.7 (0.56 to 0.84) | 1 | estimated as for clouds[8] (varied by $\pm$ 20 %) |
| $a_w$ | Albedo of water | 0.08 | 1 | from Stull (1988) |
| $\varepsilon$ | Emissivity of fiber | 0.92 | 1 | as for PVC[9] |
| $\varepsilon_r$ | Emissivity of fabric | 0.75 (0.66 to 1) | 1 | as for fiberglass[9] |
| $\varepsilon_w$ | Emissivity of water | 0.96 | 1 | from Geiger (1965) |
| $v_A$ | kinematic viscosity of air | function of $T$ | $\mathrm{cm^2 s^{-1}}$ | (Table 1) |
| $\alpha$ | Thermal diffusivity of air | function of $T$ | $\mathrm{cm^2 s^{-1}}$ | (Table 1) |
| $K_A$ | Thermal conductivity of air | function of $T$ | $\mathrm{J\,s^{-1}\,m^{-1}\,K^{-1}}$ | (Table 1) |

Sources (accessed on 3 July 2016):

1) http://www.engineeringtoolbox.com/physical-properties-thermoplastics-d_808.html

2) http://www.matweb.com/search/datasheet.aspx?MatGUID=77b5205f0dcc43bb8cbe6fee7d36cbb5

3) https://de.wikibooks.org/wiki/Tabellensammlung_Chemie/_spezifische_W%C3%A4rmekapazit%C3%A4ten

4) http://www.matweb.com/search/datasheet.aspx?matguid=8f9003366c9044bdb91bcd86e1fa6e42

5) http://www.kern.de/cgi-bin/riweta.cgi?nr=2690&lng=1

6) http://www.r-g.de/wiki/Glasfasern

7) http://www.engineeringtoolbox.com/thermal-conductivity-d_429.html

8) https://astrokramkiste.de/albedo

9) http://www.thermoworks.com/emissivity_table.html





**Table 3.** Validation of modeled temperatures by comparing with measured temperatures at 2 and 2.13 m height above the meadow and the lake respectively: Root Mean Square Errors (RMSE) for the 5–day–period from 6 to 10 April 2015. There were three different model versions: without conduction or with a lower or upper boundary for thermal conductivity $k$ of fiber and fabric.

| Compared temperatures | RMSE [K], meadow | RMSE [K], lake |
|---|---|---|
| $T_f$ (DTS) and $T_{s0}$ (model without conduction) | 0.39 | 0.25 |
| $T_f$ (DTS) and $T_{s1}$ (model with conduction, $k = 0.1 \, \mathrm{J\,s^{-1}\,m^{-1}\,K^{-1}}$) | 0.39 | 0.25 |
| $T_f$ (DTS) and $T_{s1}$ (model with conduction, $k = 0.6 \, \mathrm{J\,s^{-1}\,m^{-1}\,K^{-1}}$) | 0.40 | 0.25 |
| $T_{Af}$ (correction model) and $T_A$ (reference air temperature) | 0.42 | 0.26 |
| $T_f$ (DTS) and $T_A$ (reference air temperature) | 0.71 | 0.65 |



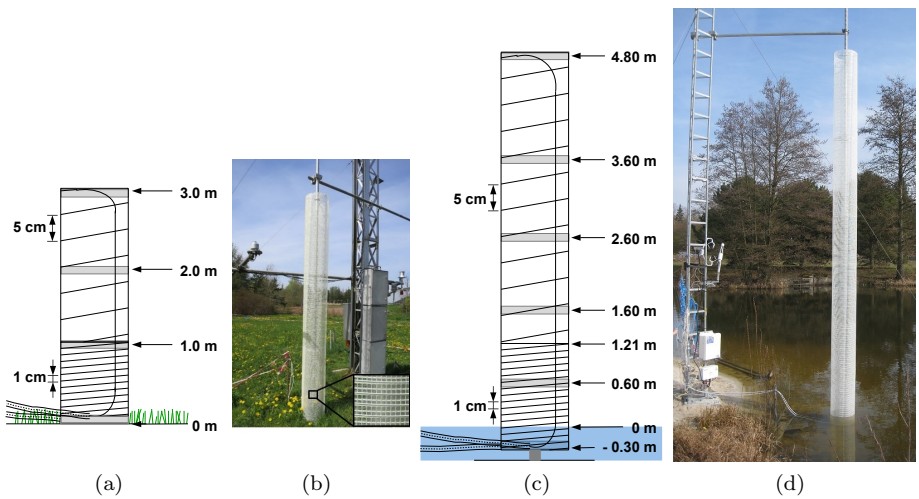

(a)    (b)    (c)    (d)

**Figure 1.** Schematic setup of the meadow–column (**a–b**) and the lake–column (**c–d**). The fiber–optic coils were spaced at 1 and 5 cm intervals in the lower and upper section respectively. Heights above ground or water surface are given for the transition of these sections and for acrylic glass rings (support structure).



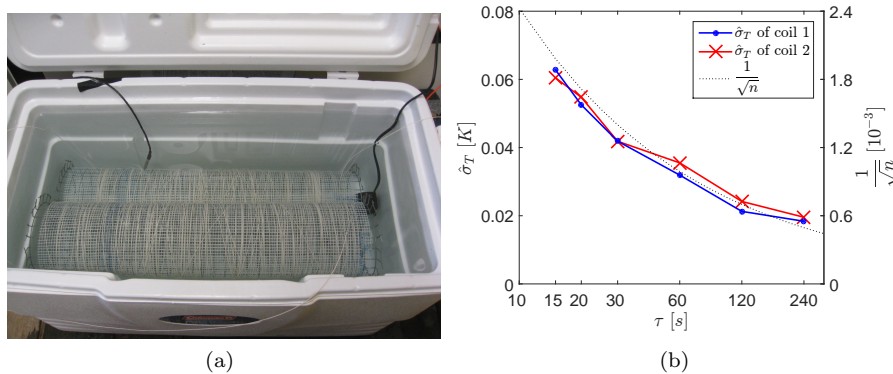

(a)  (b)

**Figure 2.** Laboratory Experiment: **a** Water bath with two fiber–optic coils, a reference thermometer (Pt100) and an aquarium pump to ensure a homogeneous water temperature. **b** Spatial standard deviation $\hat{\sigma}_T$ of measured temperatures as function of averaging time $\tau$. Depicted is the mean standard deviation of seven repetitions for each averaging time and spool of coiled fiber. For comparison the function $\sqrt{n}^{-1}$ is added with $n$ being the number of samples per double averaging time. Note: The time axis is spaced logarithmically.



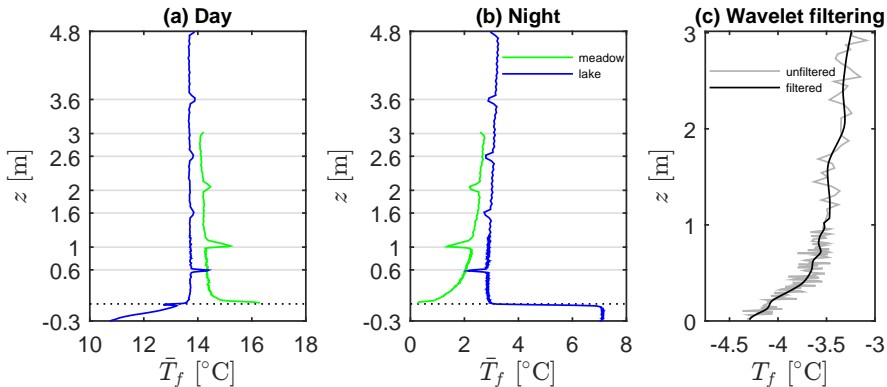

**Figure 3.** Artifacts in the measured temperature profiles: **a,b** Mean fiber temperature $\bar{T}_f$ of the meadow– and the lake–column as function of height $z$ on a sunny day with low wind velocities (10 April 2015, 06:00 – 18:00) (a) and in a night with clear sky and low wind velocities (19 – 20 March 2015, 18:00 – 06:00) (b). Systematic temperature anomalies occurred at heights where the columns were supported by acrylic glass rings. **c** Comparison between temperature profiles before and after wavelet filtering based on the BIOR5.5 set of wavelets. The critical lengths were 2.4 and 11.8 cm in the sections with higher and lower vertical resolution respectively. The unfiltered profile already consists of calibrated and 1–min averaged temperatures where artifacts from the acrylic glass rings are replaced by linear inter– or extrapolation.





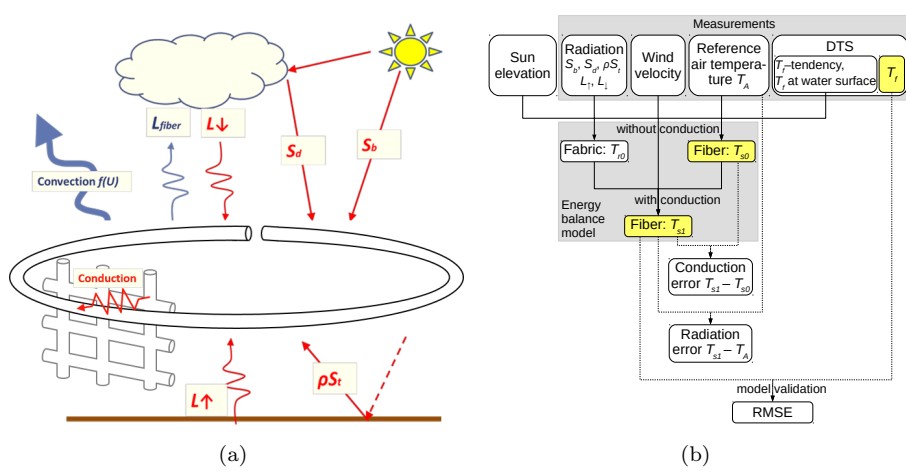

**Figure 4. a** Incoming (red) and outgoing (blue) energy balance components for a fiber coil (adapted from Sayde et al. , 2015). $S_b$, $S_d$ and $\rho S_t$ are direct, diffuse and reflected shortwave radiation respectively, $L_\downarrow$ and $L_\uparrow$ are downward and upward longwave radiation respectively and $L_{fiber}$ is longwave radiation emitted by the fiber coil. Conduction results from temperature differences between fiber coil and reinforcing fabric. **b** Flow chart of error estimation for the DTS columns with a proposed energy balance model. $T_f$ is the measured fiber temperature whereas $T_{s0}$ and $T_{s1}$ are modeled fiber temperatures. RMSE denotes Root Mean Square Error.



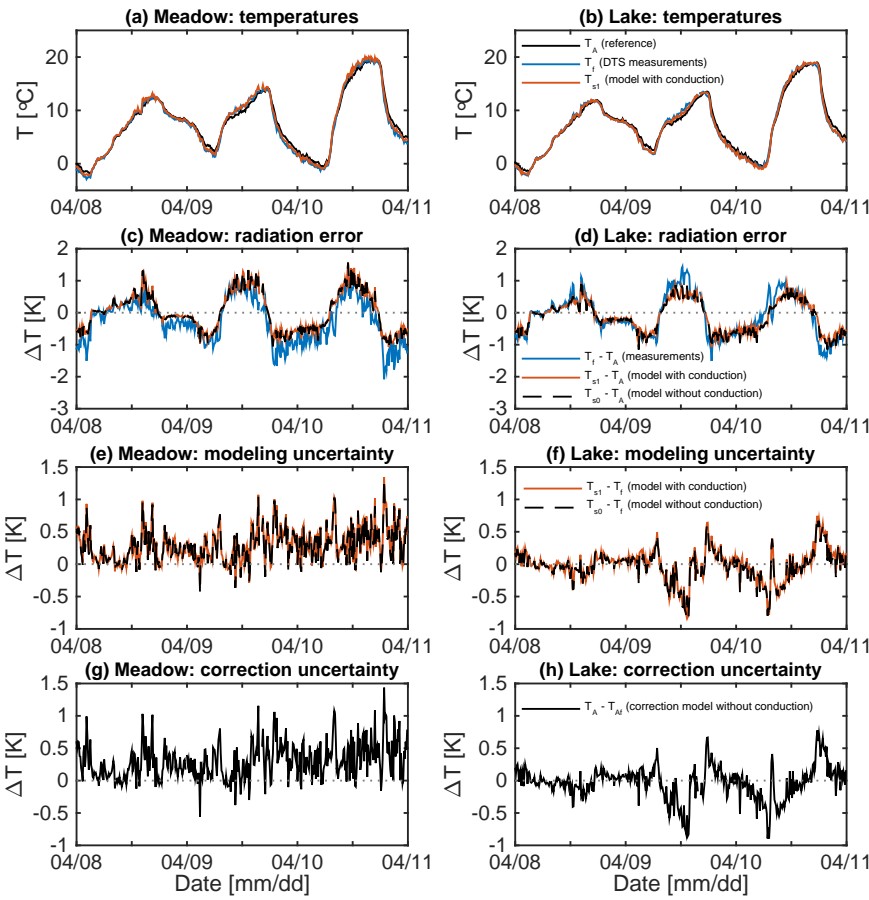

**Figure 5.** Comparison of measured and modeled temperatures from 8 to 10 April 2015 (mostly clear sky) at 2.00 and 2.13 m height above the meadow and the lake respectively: **a,b** Fiber ($T_f$, $T_{s1}$) and reference air temperatures ($T_A$). **c,d** Radiation error: fiber minus air temperature. **e,f** Modeling uncertainty: modeled minus measured fiber temperature. **g,h** Correction uncertainty: reference air minus corrected fiber temperature. In this figure $T_{s1}$ depicts fiber temperatures modeled with high thermal conductivity ($0.6\,\mathrm{J\,s^{-1}\,m^{-1}\,K^{-1}}$) but there was no difference to model versions with low conductivity or without conduction. On 9 April, 15:50 – 16:10 local standard time (LST) there was a data gap in the DTS measurements resulting from an interruption when splicing to fix a fiber break.





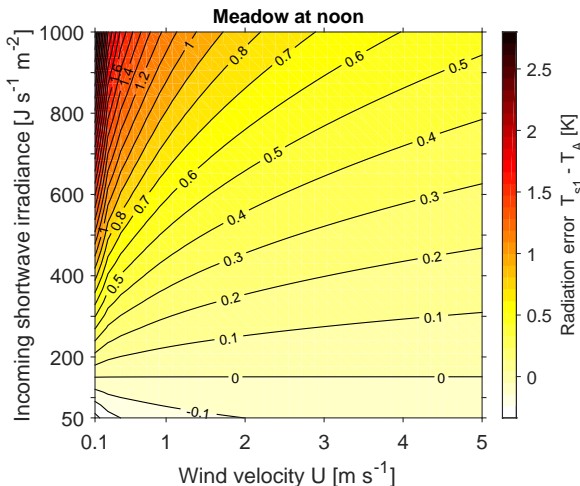

**Figure 6.** Radiation error $T_{s1} - T_A$ as function of the main drivers total incoming shortwave irradiance on a horizontal surface and wind velocity. The model predictions for 2 m height above meadow were based on conditions from 10 April 2015, 12:00 LST with an air temperature of 16.8 °C, a sun elevation of 53 °, a fixed ratio of direct to total shortwave irradiance of 0.91 and a net longwave radiation of 117 J s$^{-1}$ m$^{-2}$. Incoming shortwave irradiance and wind velocity were initially 724 J s$^{-1}$ m$^{-2}$ and 0.26 m s$^{-1}$ respectively. Reflected shortwave irradiance was calculated by multiplying the incoming shortwave irradiance with the albedo of the meadow given by the initial ratio of reflected to incoming shortwave irradiance (0.22). Here only the results for low thermal conductivity (0.1 J s$^{-1}$ m$^{-1}$ K$^{-1}$) of fabric and fiber are shown as the results for high thermal conductivity (0.6 J s$^{-1}$ m$^{-1}$ K$^{-1}$) were similar.





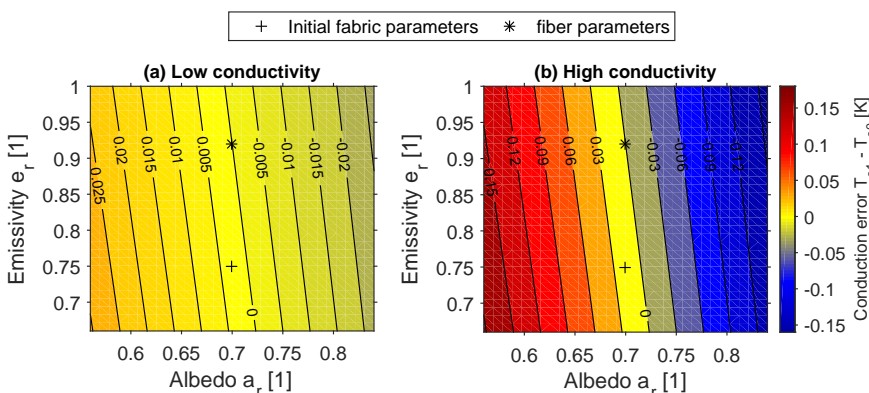

**Figure 7.** Sensitivity of the conduction error $T_{s1} - T_{s0}$ to albedo $a_r$ and emissivity $e_r$ of the reinforcing fabric at 2 m height above the meadow for conditions on 10 April 2015, 12:00 LST (incoming shortwave irradiance: 724 J s$^{-1}$ m$^{-2}$, wind velocity: 0.26 m s$^{-1}$): Model results under the assumption of a low (**a**) or a high (**b**) thermal conductivity of reinforcing fabric and fiber cable (0.1 and 0.6 J s$^{-1}$ m$^{-1}$ K$^{-1}$ respectively).



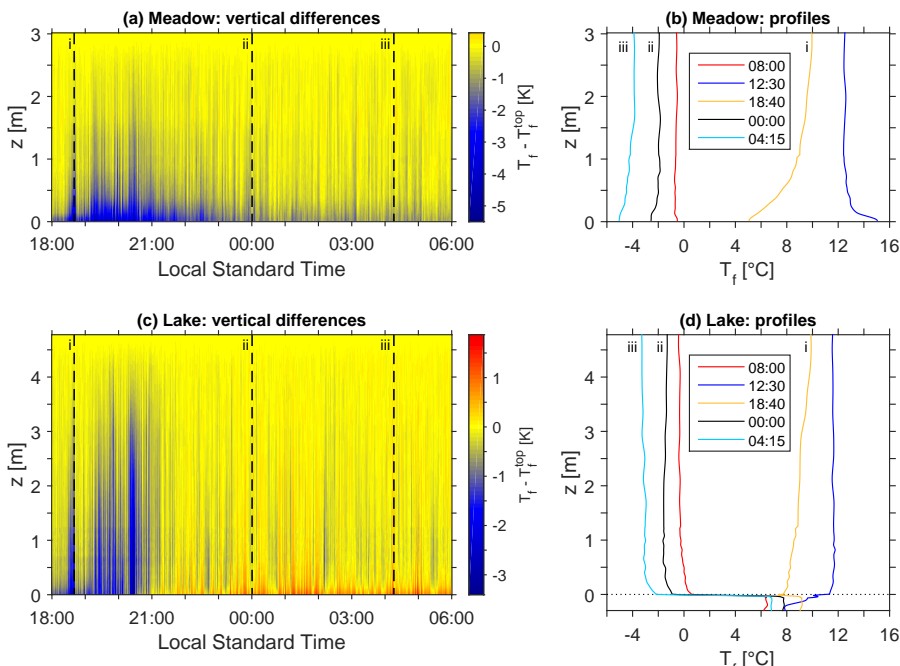

**Figure 8.** Example of DTS temperature profiles: **a,c** Vertical temperature differences $T_f - T_f^{top}$ related to the top of the profiles in the clear night from 19 to 20 March 2015. $z$ is height above ground or water surface. Water temperatures were excluded in subplot c because they differed much from the air temperatures and the color scale could otherwise not resolve the variability in the air. The section with 5 cm resolution was linearly interpolated with 1 cm resolution. The numbered dashed lines indicate points in time depicted in b and d. **b,d** Temperature profiles at selected points in time between 19 March 2015, 08:00 LST and 20 March 2015, 04:15 LST.