# Peer review of "Quantitative analysis of the radiation error for aerial coiled fiber–optic Distributed Temperature Sensing deployments using reinforcing fabric as support structure"

_Atmospheric Measurement Techniques, 2016_

## Referee Comment (RC1) · N. van de Giesen (Referee) · 23 Dec 2016

General

Nice paper that produces a very comprehensive way to correct for the effects of radiation and support structure on atmospheric DTS measurements. As such, it is a very useful addition to the rapidly growing literature on atmospheric applications of DTS. Especially the quantification of the conductivity error, and the fact that for the presented construction this turns out to be be negligible, are important.

Major comments

[Figure]

None.

Minor comments

In general, the authors seem to dislike the use of commas. Commas are not only known for their life saving properties ("Let's eat, grandma.") but they also improve readability. Especially after introductory clauses, commas are very useful. Sometimes, the authors use them there (p2 l15, p3 l1) but I would suggest to put them throughout (p2 l14, p2 l30, etc)

p2 l1: "the DTS"

p2 l16: Remove colon.

p2 l31: Remove colon.

p3 l28: There is a slight temperature dependence of Stokes as well.

p 4 l5: Should be "reinforced"?

p 4 l5: Please provide make and type of the fabric or at least a description such that we can purchase a similar fabric (material, size of skeins, size of mesh, ...).

p4 l13: Was this a tightly packed cable? Did it contain tensile strength elements such as Kevlar? The reason this is important is that later on you make the assumption that the cable is homogeneous. In addition to the actual glass fiber, were there other elements? Also, was the fiber bend-tolerant?

p 5 l19: "in the laboratory" is a bit confusing. I assume you mean that at that sampling rate, the laboratory set-up gave an error of 0.04 K.

p 5 l25: Remove colon.

p 5 l32: That seems to be very inaccurate! You are shooting from two sides so I would have expected something much better. Is there any structure in this error or is it completely random (cannot be completely random as the order of measurements

would be reversed etc). Any idea where this comes from? Did you use the speed of light assumed by the Oryx?

p8 l7: 10 should be 19

p 9 l5: Why is U assumed the same as you measure wind speed at both sites?

p9 l21: "predominantly" (interesting internet discussion on whether "predominately" is also ok, but why irk?)

p10 l10: Not sure I can agree 100%. In case of very low wind speeds, free convection will start to occur. It is not difficult to imagine that happening along the sunny side of the support structure. Eq 8 is then no longer the correct one to use. It would go too far to include this but a caution concerning free convection would be good to include here, such as "as long as free convection can be ignored."

p10 l25: It could very well be that your temperature measurement over the lake is better than over the meadow. The aspirated psychrometers may not be as great on sunny days as we often assume although one would then assume the model to underestimate the measured psychrometer values. More a loose remark than a recommendation.

p13 l2: It would be so much nicer to simply include the data set as additional material or as quotable dataset in a data depository.

---

## Short Comment (SC1) · 10 Jan 2017

Dear Nicolaas van de Giesen,

Thank you very much for your comment. The remarks are all helpful to improve the paper. Here, I want to respond to some of the remarks and questions.

I agree that a more frequent use of commas, especially after introductory clauses, will improve readability.

p4 l5: Type and manufacturer of the reinforcing fabric were the following and will be included in the manuscript: maxit Armierungsgewebe PS, maxit Baustoffwerke GmbH,

Krölpa, TH, Germany.

p4 l13: The used fiber–optic cable was composed of a 50 μm glass core, a 75 μm thick glass cladding, Kevlar fibers for tensile strength, and a polyvinyl chloride (PVC) coating with 900 μm outer diameter. This cable was tightly buffered and bend-insensitive. Since the used energy balance model was based on the assumption of a homogeneous cable, the material properties of the cable were roughly estimated using the arithmetic mean for the three components PVC, Kevlar fiber and glass (Table 2).

p5 l32: The accuracy of $\pm 2$ m along the fiber refers to the transformation of the measurement positions from length along the fiber into height above surface, i.e. the measurement used for a certain height could have been separated by 2 m fiber length from the actual measurement at that height. This accuracy was indicated by cross–checking the obtained heights against the counted fiber coils. For the lower part of the temperature profiles, $\pm 2$ m along the fiber corresponded to $\pm 2$ cm in height. The accuracy was not meant to be the accuracy of the positions in length along the fiber. The length along the fiber was provided by the Oryx and based on the speed of light. I realize that the sentence in the manuscript can be misunderstood and needs to be written more clearly.

p8 l19: I agree that Eq. 7 contains a mistake. 10 should be 19.

p9 l3: The wind speed measured at 2.13 m height above the lake was also used for 2.0 m height at the meadow–column because wind speed was only measured at 17 m height at the meadow site.

p10 l10: Do you mean that the parameterization of the convection heat transfer coefficient (Eq. 8) might not always be applicable because free convection could happen only along the sunny side of the support structure which would not be captured by a single, nearby sonic anemometer? This might be true but this case can be assumed to happen seldom.

p13 l2: We, the authors, consider making the dataset available via a data depository. Would you prefer raw DTS data or final corrected DTS data?

---

## Referee Comment (RC2) · Anonymous Referee #3 · 12 Mar 2017

The paper describes analysis of radiation error of a specific implementation of fiber-optic Raman scatter DTS aimed at profiling temperature in the atmospheric surface layer. The DTS system with coiled glass fiber is supported with meshed fabric to keep the shape and mounted on a small masts of ∼3m and ∼5m heights to profile air temperature in two locations over ground and water. Radiative heating and cooling of such structure is non-negligible, thus, in order to measure temperature of the air corrections for these effects are necessary. Detailed analysis of these corrections is the core of the paper. The authors developed and tested a simple energy balance model of the measurement system. This model allowed to estimate temperature corrections. The

temperatures retrieved after correction (using two slightly different models accounting or not accounting for heat conduction) were compared to air temperatures measured by reference thermometers at 2m height indicating a reasonable agreement. The prosed approach is a step forward in temperature profiling the atmospheric surface layer and the paper is suitable for AMT, but requires enhanced discussion before final acceptance.

Remarks.

1) Discuss, please, briefly, time response to variations in solar radiation and wind velocity – might be important for variable solar fluxes due to cloudiness and fluctuating wind speeds. Is it possible that some of the noise in Ta-Tf results from effectively different filtering (averaging) of Ta and Tf?

2) Fig. 5 documents a systematic offset between DTS and reference temperatures over a meadow and underestimate of an amplitude of correction over a lake. Behavior of the proposed corrections seems to be dependent on the localization of the measurement site. Discuss more, please, taking into account different reference thermometers in both localizations. Do you have a crude idea which terms in the energy balance model contribute mostly to these differences?

---

## Author Comment (AC1) · 7 Apr 2017

Our responses are marked in *italic and blue* and were directly inserted below each comment.

**1. Review comments and author's response**

*We thank both referees for their constructive review comments which helped to further improve the manuscript.*

[Figure]

**1.1. Comments from Nicolaas van de Giesen**

General

Nice paper that produces a very comprehensive way to correct for the effects of radiation and support structure on atmospheric DTS measurements. As such, it is a very useful addition to the rapidly growing literature on atmospheric applications of DTS. Especially the quantification of the conductivity error, and the fact that for the presented construction this turns out to be be negligible, are important.

Major comments

None.

Minor comments

In general, the authors seem to dislike the use of commas. Commas are not only known for their life saving properties ("Let's eat, grandma.") but they also improve readability. Especially after introductory clauses, commas are very useful. Sometimes, the authors use them there (p2 l15, p3 l1) but I would suggest to put them throughout (p2 l14, p2 l30, etc)

*In order to improve readability, we have inserted much more commas throughout the paper, especially after introductory clauses.*

p2 l1: "the DTS"

*We have corrected this write error.*

p2 l16: Remove colon.

*We have replaced the colon by a dot.*

p2 l31: Remove colon.

*We have replaced the colon by a dot.*

p3 l28: There is a slight temperature dependence of Stokes as well.

*We agree that the Stokes signal slightly depends on fiber temperature, but much less than the anti–Stokes signal (van de Giesen et al., 2012). We have restated the sentence in our manuscript.*

p 4 l5: Should be "reinforced"?

*With "reinforcing fabric column" we meant a column made of reinforcing fabric, which is now restated.*

p 4 l5: Please provide make and type of the fabric or at least a description such that we can purchase a similar fabric (material, size of skeins, size of mesh, ...).

*Type and manufacturer of the reinforcing fabric were the following and have been included on page 4, line 5:*
*maxit Armierungsgewebe PS, maxit Baustoffwerke GmbH, Krölpa, TH, Germany.*

p4 l13: Was this a tightly packed cable? Did it contain tensile strength elements such as Kevlar? The reason this is important is that later on you make the assumption that the cable is homogeneous. In addition to the actual glass fiber, were there other elements? Also, was the fiber bend–tolerant?

*The used fiber–optic cable was tightly buffered and bend–insensitive. This cable was composed of a 50 µm glass core, a glass cladding with 125 µm diameter, Kevlar fibers embedded in a buffer, and a polyvinyl chloride (PVC) jacket with 900 µm outer diameter. Since the used energy balance model was based on the assumption of a homogeneous cable, the material properties of the cable were roughly estimated using the arithmetic mean for the three components PVC, Kevlar fiber and glass (Table 2).*

p 5 l19: "in the laboratory" is a bit confusing. I assume you mean that at that sampling rate, the laboratory set–up gave an error of 0.04 K.

*Yes, that is correct. Now, the sentence is more precise.*

p 5 l25: Remove colon.

*We have replaced the colon by a dot.*

p 5 l32: That seems to be very inaccurate! You are shooting from two sides so I would have expected something much better. Is there any structure in this error or is it completely random (cannot be completely random as the order of measurements would be reversed etc). Any idea where this comes from? Did you use the speed of light assumed by the Oryx?

*The Oryx reported measurement positions in length along the fiber based on the speed of light, which is accurate. In order to compute the measurement heights above the surface, we attached cold packs to individual fiber sections at known heights and thus created a temperature minimum at respective lengths along the fiber. Some uncertainty resulted from the fact that the temperatures of two subsequent sampling intervals along the fiber are not completely independent from each other and consequently the created temperature minimum was not always defined sharply. The computed heights above surface were cross–checked by counting the number of fiber coils for each column. The chross–check indicated that the computed heights had an accuracy of approximately $\pm 2$ cm and $\pm 10$ cm for the lower and upper parts of the columns, respectively, because the fiber length between the lowest and uppermost computed heights differed by maximum 2 m from the counted fiber length. Although the actual measurement heights seemed to be inaccurate, the heights were accurate relative to each other.*

*The precision of the temperature signal depended on the direction of the DTS measurement. Due to the double–ended DTS configuration, the temperature profile had alternately a higher and lower precision because the columns were alternately located*

*at the beginning and end of the overall measurement path. For this reason, subsequent measurements were averaged, which was done after transforming the measurement positions from length along the fiber into height above surface.*

p8 l7: 10 should be 19

*Yes, this was a mistake. We have replaced 10 by 19 in Eq. 7.*

p 9 l5: Why is U assumed the same as you measure wind speed at both sites?

*The wind speed measured at 2.13 m height above the lake was also used for 2.0 m height at the meadow–column because wind speed was only measured at 17 m height at the meadow site.*

p9 l21: "predominantly" (interesting internet discussion on whether "predominately" is also ok, but why irk?)

*We have replaced "predominately" by "predominantly".*

p10 l10: Not sure I can agree 100 %. In case of very low wind speeds, free convection will start to occur. It is not difficult to imagine that happening along the sunny side of the support structure. Eq 8 is then no longer the correct one to use. It would go too far to include this but a caution concerning free convection would be good to include here, such as "as long as free convection can be ignored."

*The parameterization of the convection heat transfer coefficient (Eq. (8)) can be used for a wide range of Reynolds numbers from $1$ to $10^6$ (Zkauskas, 1987). Since the onset of free convection is difficult to determine and we cannot guarantee that Eq. (8) is valid in the case of free convection, we have included the proposed caution "as long as free convection can be ignored". In our experience, however, the parameterization provided reasonable results also for very unstable conditions.*

p10 l25: It could very well be that your temperature measurement over the lake is better

than over the meadow. The aspirated psychrometers may not be as great on sunny days as we often assume although one would then assume the model to underestimate the measured psychrometer values. More a loose remark than a recommendation.

*At the meadow site, the corrected fiber temperature seemed to systematically under-estimate the measured psychrometer values (Fig. 5g,h). Since this applied to both day and night, we do not think that the mismatch over the meadow is caused by the radia-tion error of the aspirated psychrometer. Instead, the mismatch could, at least partly, result from the fact that the DTS temperature at 2.00 m height above the meadow was linearly interpolated because of the presence of an acrylic glass ring at that height, which introduced some additional uncertainty compared to the lake site.*

p13 l2: It would be so much nicer to simply include the data set as additional material or as quotable dataset in a data depository.

*We can make the data available using a data depository. Would you prefer raw DTS data or final corrected DTS data?*

**1.2. Comments from Referee #3**

The paper describes analysis of radiation error of a specific implementation of fiber–optic Raman scatter DTS aimed at profiling temperature in the atmospheric surface layer. The DTS system with coiled glass fiber is supported with meshed fabric to keep the shape and mounted on a small masts of ~3m and ~5m heights to profile air tem-perature in two locations over ground and water. Radiative heating and cooling of such structure is non–negligible, thus, in order to measure temperature of the air corrections for these effects are necessary. Detailed analysis of these corrections is the core of the paper. The authors developed and tested a simple energy balance model of the measurement system. This model allowed to estimate temperature corrections. The temperatures retrieved after correction (using two slightly different models accounting

or not accounting for heat conduction) were compared to air temperatures measured by reference thermometers at 2m height indicating a reasonable agreement. The prosed approach is a step forward in temperature profiling the atmospheric surface layer and the paper is suitable for AMT, but requires enhanced discussion before final acceptance.

Remarks.

1) Discuss, please, briefly, time response to variations in solar radiation and wind velocity – might be important for variable solar fluxes due to cloudiness and fluctuating wind speeds. Is it possible that some of the noise in Ta-Tf results from effectively different filtering (averaging) of Ta and Tf?

*The energy balance model was run on an 10 min basis. For both the reference temperature $T_A$ and the DTS temperature $T_f$, 10 min averages are shown in Fig. 5. In comparison, the conductive time scale of the radial heat transfer within the fiber–optic cable is approximately 1.1 s (Thomas et al., 2012), i.e. temperature changes due to variations in solar radiation and wind speeds are captured quickly. For the reference thermometers, the response time is also smaller than the averaging time of 10 min. Therefore, an effectively different filtering (averaging) of $T_A$ and $T_f$ is not responsible for the noise in $T_A - T_f$. Since this noise was mainly observed over the meadow, it could result from the fact that the DTS temperature at 2.00 m height above the meadow was linearly interpolated because of the presence of an acrylic glass ring at that height.*

2) Fig. 5 documents a systematic offset between DTS and reference temperatures over a meadow and underestimate of an amplitude of correction over a lake. Behavior of the proposed corrections seems to be dependent on the localization of the measurement site. Discuss more, please, taking into account different reference thermometers in both localizations. Do you have a crude idea which terms in the energy balance model contribute mostly to these differences?
*On 8 April, a relatively good agreement between modeled and measured temperatures compared to the subsequent two days was found, especially above the lake (Fig. 5f,h). On 8 April, the sky was covered by more clouds compared to the other days. Thus, the energy balance model may be sensitive to uncertainties in the terms for shortwave and longwave radiation. In contrast to the meadow site, the outgoing longwave irradiance was not directly measured at the lake site but parameterized via the Stefan–Boltzmann law. Uncertainties in estimating the water surface temperature from the DTS measurements may substantially contribute to temporary under- or overestimation of the radiation error at the lake, especially in the non–stationary transition periods between day and night and for little wind–induced mixing of the water. The lower accuracy of the reference temperatures from the ultrasonic anemometer over the lake compared to the aspirated psychrometer over the meadow could also play a role. However, this would rather cause an offset than temporary under- or overestimation of the radiation error at the lake.*

*Over the meadow, the model seemed to systematically overestimate fiber temperatures although the aspirated psychrometer provided accurate reference temperatures and we corrected for an instrument–specific offset between DTS and reference temperatures. The overestimation of the fiber temperature could result from an underestimation of the fiber emissivity, which would especially reduce the emission of longwave radiation, and an underestimation of the fiber albedo, which would enhance the absorption of shortwave radiation. However, this does not seem to be the only reason because a similar systematic offset could then also be expected over the lake, which was not the case. The uncertainties resulting from the assumption of an identical wind velocity at both locations and from linear interpolation of the DTS temperature at 2.00 m height above the meadow because of the presence of an acrylic glass ring at that height, could also contribute to the offset between model and measurement over the meadow.*

**2. Author's changes in the manuscript**

*We have adopted the suggestions of the referee comments or restated respective sentences and paragraphs in order to be more precise. A marked–up manuscript version is supplemented to this author comment.*

[revised manuscript text omitted]

The post-processing also included the transformation of the measurement positions from length along the fiber into height above surface. For mapping purposes this purpose, cold packs were attached to individual sections of the fiber were cooled at

- 5 the beginning and end of each columnat known heights, while the positions heights in between were inferred by means of the column proportions. Results-The computed heights were cross-checked against the counted number of fiber coils. The This cross-check indicated an accuracy of the positions was computed heights of approximately  $\pm 2$  m along the fiber corresponding to  $\pm 2$  and  $\pm 10$  cm in height -for the lower and upper parts of the columns, respectively. Nevertheless, the obtained heights were accurate relative to each other.

[revised manuscript text omitted]

(6)

25 where  $\bar{S}_{b,cyl}$  [Js-1m-2] is the mean direct radiation over unit cylinder surface area,  $S_{b,h}$  [Js-1m-2] is direct radiation measured on a horizontal plane,  $\theta$  [°] is the angle between cylinder axis and solar azimuth and  $\beta$  [°] is sun elevation which was calculated from the geographic coordinates and time by means of a flux data processing software. A fiber coil can be approximated by several cylinder segments with varying orientation, i.e. varying angles  $\theta$ . Hence, the mean direct radiation  $\bar{S}_b$ [Js-1m-2] intercepted by a fiber coil was calculated by averaging  $\bar{S}_{b,cyl}$  for 19 different angles  $\theta$ :

30
$$\bar{S}_b = \frac{1}{10} \frac{1}{19} \sum_{i=0}^{18} \bar{S}_{b,cyl} (\theta = 5^\circ \cdot i)$$
 (7)

A number of 19 angles  $\theta$  was deemed a sufficient approximation as a doubling of this number would change  $\bar{S}_b$  by less than 0.6 %.

The convective heat–transfer coefficient h from Eq. (4) mainly depends on wind velocity and can be parameterized as (Sayde et al., 2015; Lundström et al., 2007; Zkauskas, 1987):

5
$$h = C (2r)^{m-1} \operatorname{Pr}^n \left( \frac{\operatorname{Pr}}{\operatorname{Pr}_s} \right)^{\frac{1}{4}} K_A v_A^{-m} U_N^m$$
(8)

where r [m] is fiber cable radius,  $K_A$  [J s-1 m-1 K-1] is thermal conductivity of air and  $v_A$  [m2 s-1] is kinematic viscosity of air. The dimensionless parameters C and m were set 0.52 and 0.5 respectively for Reynolds numbers from 40 to 1000 and they were set 0.75 and 0.4 respectively for Reynolds numbers between 1 and 40 (Zkauskas, 1987). Pr and Prs [1] are Prandtl numbers at air and fiber temperature respectively. The Prandtl numbers are given by  $v \alpha^{-1}$  and were calculated by linear inter-

[revised manuscript text omitted]

| $\upsilon  [\mathrm{cm}^2  \mathrm{s}^{-1}]$ | $K  [\mathrm{J  s^{-1}  m^{-1}  K^{-1}}]$ | $\alpha  [\rm cm^2  \rm s^{-1}]$                                                                                                                                                                    |
|----------------------------------------------|-------------------------------------------|-----------------------------------------------------------------------------------------------------------------------------------------------------------------------------------------------------|
| 0.132                                        | 0.024                                     | 0.184                                                                                                                                                                                               |
| 0.141                                        | 0.025                                     | 0.196                                                                                                                                                                                               |
| 0.150                                        | 0.025                                     | 0.208                                                                                                                                                                                               |
| 0.160                                        | 0.026                                     | 0.233                                                                                                                                                                                               |
|                                              | $v [cm^2 s^{-1}]$ 0.132 0.141 0.150 0.160 | $\begin{array}{c} \upsilon  [{\rm cm}^2  {\rm s}^{-1}] & K  [{\rm J}  {\rm s}^{-1}  {\rm m}^{-1}  {\rm K}^{-1}] \\ \\ 0.132 & 0.024 \\ 0.141 & 0.025 \\ 0.150 & 0.025 \\ 0.160 & 0.026 \end{array}$ |

| Symbol          | Description                      | Values          | Units                          | Explanation                                                                   |
|-----------------|----------------------------------|-----------------|--------------------------------|-------------------------------------------------------------------------------|
| $c_p$           | Specific heat capacity of fiber  | 1.06            | $\rm kJkg^{-1}K^{-1}$          | mean for $PVC^{1}$ , Kevlar fiber 2 )
and glass 3    |
| $c_{pr}$        | Specific heat capacity of fabric | 0.80            | $\rm kJkg^{-1}K^{-1}$          | as for A–glass fiber 4)                                            |
| ρ               | Density of fiber                 | 1.8             | ${ m gcm^{-3}}$                | mean for $PVC^{5)}$ , Kevlar fiber 2)
and E-glass 6) |
| $ ho_r$         | Density of fabric                | 2.4             | ${ m gcm^{-3}}$                | as for A–glass fiber 4)                                            |
| r               | Radius of fiber or fabric        | 0.45            | mm                             |                                                                               |
| В               | Length of fiber or fabric        | 1.01            | m                              |                                                                               |
| k               | Thermal conductivity for         | low: 0.1        | $\rm Js^{-1}m^{-1}K^{-1}$      | mean for $PVC^{5)}$ (fiber                                                    |
|                 | fiber and fabric                 | high: 0.6       |                                | jacket) and fiberglass/ glass 7)                                   |
|                 |                                  |                 |                                | (fabric)                                                                      |
| A               | Contact area of fiber and fabric | 0.5             | $\mathrm{cm}^2$                | estimated                                                                     |
| a               | Albedo of fiber                  | 0.7             | 1                              | estimated as for clouds 8)                                         |
| $a_r$           | Albedo of fabric                 | 0.7             | 1                              | estimated as for clouds 8)                                         |
|                 |                                  | (0.56 to 0.84)  |                                | (varied by $\pm~20~\%)$                                                       |
| $a_w$           | Albedo of water                  | 0.08            | 1                              | from Stull (1988)                                                             |
| ε               | Emissivity of fiber              | 0.92            | 1                              | as for PVC 9)                                                      |
| $\varepsilon_r$ | Emissivity of fabric             | 0.75            | 1                              | as for fiberglass 9)                                               |
|                 |                                  | (0.66 to 1)     |                                |                                                                               |
| $\varepsilon_w$ | Emissivity of water              | 0.96            | 1                              | from Geiger (1965)                                                            |
| $v_A$           | kinematic viscosity of air       | function of $T$ | $\mathrm{cm}^2\mathrm{s}^{-1}$ | (Table 1)                                                                     |
| $\alpha$        | Thermal diffusivity of air       | function of $T$ | $\mathrm{cm}^2\mathrm{s}^{-1}$ | (Table 1)                                                                     |
| $K_A$           | Thermal conductivity of air      | function of $T$ | $\rm Js^{-1}m^{-1}K^{-1}$      | (Table 1)                                                                     |

**Table 2.** Material properties used in the energy balance model. A fiber cable coil and a reinforcing fabric skein are briefly called fiber and fabric respectively. Numbers in brackets express the range used for sensitivity analysis. *T* denotes temperature.

Sources (accessed on 3 July 2016):

1) http://www.engineeringtoolbox.com/physical-properties-thermoplastics-d\_808.html

2) http://www.matweb.com/search/datasheet.aspx?MatGUID=77b5205f0dcc43bb8cbe6fee7d36cbb5

3) https://de.wikibooks.org/wiki/Tabellensammlung\_Chemie/\_spezifische\_W%C3%A4rmekapazit%C3%A4ten

4) http://www.matweb.com/search/datasheet.aspx?matguid = 8f9003366c9044bdb91bcd86e1fa6e42

5) http://www.kern.de/cgi-bin/riweta.cgi?nr=2690&lng=1

6) http://www.r-g.de/wiki/Glasfasern

7) http://www.engineeringtoolbox.com/thermal-conductivity-d\_429.html

8) https://astrokramkiste.de/albedo

9) http://www.thermoworks.com/emissivity\_table.html

**Table 3.** Validation of modeled temperatures by comparing with measured temperatures at 2 and 2.13 m height above the meadow and the lake respectively: Root Mean Square Errors (RMSE) for the 5–day–period from 6 to 10 April 2015. There were three different model versions: without conduction or with a lower or upper boundary for thermal conductivity k of fiber and fabric.

| Compared temperatures                                             | RMSE [K], meadow | RMSE [K], lake |
|-------------------------------------------------------------------|------------------|----------------|
| $T_f$ (DTS) and $T_{s0}$ (model without conduction)               | 0.39             | 0.25           |
| $T_f$ (DTS) and $T_{s1}$ (model with conduction,                  |                  |                |
| $_{\sim}k = 0.1 \text{ J s}^{-1} \text{ m}^{-1} \text{ K}^{-1}$ ) | 0.39             | 0.25           |
| $T_f$ (DTS) and $T_{s1}$ (model with conduction,                  |                  |                |
| $_{\sim}k = 0.6 \text{ J s}^{-1} \text{ m}^{-1} \text{ K}^{-1}$ ) | 0.40             | 0.25           |
| $T_{Af}$ (correction model) and $T_A$ (reference air temperature) | 0.42             | 0.26           |
| $T_f$ (DTS) and $T_A$ (reference air temperature)                 | 0.71             | 0.65           |

Figure 1. Schematic setup of the meadow-column (a-b) and the lake-column (c-d). The fiber-optic coils were spaced at 1 and 5 cm intervals in the lower and upper section respectively. Heights above ground or water surface are given for the transition of these sections and for acrylic glass rings (support structure).